# Testing and healthcare seeking behavior preceding HIV diagnosis among migrant and non-migrant individuals living in the Netherlands: Directions for early-case finding

Ward P. H. van Bilsen[1]*, Janneke P. Bil[1], Jan M. Prins[2], Kees Brinkman[3], Eliane Leyten[4], Ard van Sighem[5], Maarten Bedert[1], Udi Davidovich[1,6], Fiona Burns[7], Maria Prins[1,2]

1 Department of Infectious Diseases, Public Health Service of Amsterdam, Amsterdam, The Netherlands, 2 Department of Internal Medicine, Division of Infectious Diseases, Amsterdam UMC, University of Amsterdam, Amsterdam, The Netherlands, 3 Department of Internal Medicine, Division of Infectious Diseases, OLVG, Amsterdam, The Netherlands, 4 Department of Internal Medicine, Division of Infectious Diseases, Haaglanden Medical Center, Den Haag, The Netherlands, 5 Stichting HIV Monitoring, Amsterdam, The Netherlands, 6 Department of Social Psychology, University of Amsterdam, Amsterdam, The Netherlands, 7 Insitute for Global Health, University College London, London, United Kingdom

* wvbilsen@ggd.amsterdam.nl

**Data Availability Statement:** The Medical Ethical Committee of the University of Amsterdam put restrictions on publicly sharing data of the aMASE

## Abstract

### Objectives

To assess differences in socio-demographics, HIV testing and healthcare seeking behavior between individuals diagnosed late and those diagnosed early after HIV-acquisition.

### Design

Cross-sectional study among recently HIV-diagnosed migrant and non-migrant individuals living in the Netherlands.

### Methods

Participants self-completed a questionnaire on socio-demographics, HIV-testing and health-care seeking behavior preceding HIV diagnosis between 2013–2015. Using multivariable logistic regression, socio-demographic determinants of late diagnosis were explored. Variables on HIV-infection, testing and access to care preceding HIV diagnosis were compared between those diagnosed early and those diagnosed late using descriptive statistics.

### Results

We included 143 individuals with early and 101 with late diagnosis, of whom respectively 59/143 (41%) and 54/101 (53%) were migrants. Late diagnosis was significantly associated with older age and being heterosexual. Before HIV diagnosis, 89% of those with early and 62% of those with late diagnosis had ever been tested for HIV-infection ($p<0.001$), and respectively 99% and 97% reported healthcare usage in the Netherlands in the two years preceding HIV diagnosis ($p = 0.79$). Individuals diagnosed late most frequently visited a

study as it contains potentially identifying information of human subjects. The data that support the findings of this study are however available upon reasonable request from the corresponding author (wvbilsen@ggd.amsterdam.nl) or the aMASE study group (amatser@ggd.amsterdam.nl). Data requests will be reviewed by the aMASE study group.

**Funding:** The author(s) received no specific funding for this work.

**Competing interests:** The authors have declared that no competing interests exist.

general practitioner (72%) or dentist (62%), and 20% had been hospitalized preceding diagnosis. In these settings, only in respectively 20%, 2%, and 6% HIV-testing was discussed.

## Conclusion

A large proportion of people diagnosed late had previously tested for HIV and had high levels of healthcare usage. For earlier-case finding of HIV it therefore seems feasible to successfully roll out interventions within the existing healthcare system. Simultaneously, efforts should be made to encourage future repeated or routine HIV testing among individuals whenever they undergo an HIV test.

## Introduction

In the Netherlands, the number of new HIV diagnoses decreased from 1,026 in 2013 to 580 in 2019. The majority of diagnoses were among men who reported sex with other men (61%) and a relatively large proportion was born outside the Netherlands (42%) [1]. Over the past years, efforts have been made to increase HIV testing uptake among specific subgroups. Individuals at higher risk, e.g. men who have sex with men (MSM) and sex workers, are offered free-of-charge HIV testing at Sexual Health Centers (SHC) of Public Health Services and during outreach activities. Individuals can also test for HIV at the general practitioner (GP), which costs approximately €10 as part of the deductible excess of one's health insurance. Individuals without a health insurance or residence permit are also able to get tested for HIV at the GP or SHC.

Despite wide-spread availability of HIV tests in the Netherlands, still one out of every three individuals newly diagnosed with HIV received their diagnosis at a late stage of disease [1]. Late HIV diagnoses, i.e. having $<350/mm^3$ $CD4^+$-cells or an AIDS-defining illness at time of HIV diagnosis [2], have been defined as a major public health challenge in ending the HIV epidemic, as it contributes to ongoing HIV transmission [3]. On the individual level, late diagnosis is associated with adverse clinical outcomes [4, 5]. It is therefore important to identify and treat individuals with an HIV-infection as early as possible. For the development of effective interventions to increase earlier diagnosis, more knowledge is needed about factors associated with late diagnosis. Also, it is important to assess which opportunities for testing are currently being missed. In this study we aimed to assess whether socio-demographic characteristics, HIV testing and healthcare seeking behavior preceding HIV diagnosis differed between individuals with early and late diagnosis in the Netherlands.

## Methods

### Study design and population

Data was used from individuals living in the Netherlands who participated in the European advancing Migrant Access to health services in Europe (aMASE) study. Inclusion criteria and study procedures of the aMASE study are described elsewhere [6, 7]. Briefly, migrant and non-migrant individuals aged ≥18 years with an HIV diagnosis in the five years preceding recruitment were included between 2013–2015. Recruitment in the Netherlands was conducted at three HIV outpatient treatment clinics in Amsterdam and The Hague. Participants completed a self-administered computer-assisted questionnaire or personal interview on socio-demographics, and use of HIV-related services before and after HIV diagnosis (see S1 and S2

Appendices). Questionnaire items were mostly generated using existing survey instruments, and new questions were drafted by the aMASE research team and international experts and EuroCoord calloborators [6]. Validation of the questionnaire was done using cognitive testing by Latin American and black African migrants living in Spain and England who were recruited from community-based HIV service organizations [8]. Questionnaire items were available in 15 languages. Clinical data were obtained from the ATHENA (AIDS Therapy Evaluation in the Netherlands) HIV cohort database [1].

## Study variables

Socio-demographic characteristics included age at study inclusion and HIV diagnosis, gender, sexual orientation, education level, income, migration background (defined as born outside the Netherlands), region of birth, self-defined ethnicity, injecting drug use and attending religious services at least once a year.

Variables related to HIV diagnosis included years since HIV diagnosis at study inclusion, location of HIV diagnosis, and $CD4^+$-cell count and AIDS-defining illness at time of HIV diagnosis. Late HIV diagnoses was defined as having $<350/mm^3$ $CD4^+$-cells or an AIDS-defining illness at time of HIV diagnosis [2]. Among migrants, years between migration and HIV diagnosis and country of HIV diagnosis was additionally assessed. Variables on testing behavior preceding HIV diagnosis included ever having had an HIV test before diagnosis, and, among those with a previous HIV test, years between last negative HIV test and HIV diagnosis. Self-reported diagnosis of hepatitis B virus (HBV), hepatitis C virus (HCV) or bacterial sexually-transmitted infection (STI) preceding HIV diagnosis was additionally assessed.

Variables on access to healthcare preceding HIV diagnosis included registration at a GP in the Netherlands at time of HIV diagnosis, healthcare usage in the Netherlands in the two years preceding HIV diagnosis, and experienced difficulty accessing healthcare in the Netherlands. Among those with healthcare usage in the two years preceding HIV diagnosis, we assessed which healthcare professionals were visited and whether an HIV test was discussed during these visits.

## Statistical analyses

First, we constructed a multivariable logistic regression model to explore whether socio-demographic characteristics were associated with late presentation. Variables with a $p$-value$<0.2$ in univariable analysis were included in a multivariable model, after which all non-significant variables were subsequently removed in a backwards-stepwise fashion. Second, variables on HIV diagnosis, and HIV testing and access to care preceding HIV diagnosis were compared between those with an early and late HIV diagnosis using Pearson's $\chi^2$ test or Fisher's exact test for categorical data and Mann-Whitney U test for continuous data. As sexual orientation and migrant background was shown to affect previous outcomes of aMASE studies [6, 9], participants were categorized in the following six groups: non-migrant men who have sex with men (MSM), migrant MSM, non-migrant heterosexual men, migrant heterosexual men, non-migrant women and migrant women.

All analyses were conducted in STATA IC v15.0. A $p$-value$<0.05$ was considered statistically significant.

## Ethical considerations

The medical ethical committee of the University of Amsterdam approved the aMASE study in the Netherlands (2013_137#C20131038). Informed consent was obtained for all participants through a tick box in the questionnaire.

**Table 1. Socio-demographic characteristics of aMASE-study participants in the Netherlands, 2013–2015, including uni- and multivariable logistic regression analysis of determinants associated with late diagnosis of HIV infection.**

| | Early diagnosis* (N = 143) | | Late diagnosis* (N = 101) | | Association with late diagnosis*, univariable analyses | | | Association with late diagnosis *, multivariable analysis | | |
|---|---|---|---|---|---|---|---|---|---|---|
| | n/N | % | n/N | % | OR | 95%-CI | p-value | aOR | 95%-CI | p-value |
| **Age at HIV-diagnosis (median, IQR)** | 37 | [29–48] | 44 | [37–50] | 1.04 | 1.01–1.06 | .001 | 1.04 | 1.01–1.06 | .007 |
| **Male** | 130/143 | 91% | 90/101 | 89% | 0.82 | 0.35–1.91 | .643 | | | |
| **Sexual orientation according to migration status** | | | | | | | | | | |
| Non-migrant MSM | 75/143 | 53% | 37/101 | 37% | Ref | | < .001 | Ref | | < .001 |
| Migrant MSM | 48/143 | 34% | 26/101 | 26% | 1.10 | 0.59–2.04 | | 1.34 | 0.70–2.57 | |
| Non-migrant heterosexual male | 4/143 | 3% | 10/101 | 10% | 5.07 | 1.49–17.24 | | 5.57 | 1.60–19.40 | |
| Migrant heterosexual male | 3/143 | 2% | 17/101 | 17% | 11.48 | 3.17–41.69 | | 11.14 | 3.04–40.81 | |
| Migrant heterosexual female | 8/143 | 6% | 11/101 | 11% | 2.79 | 1.03–7.52 | | 3.20 | 1.16–8.88 | |
| Non-migrant heterosexual female | 5/143 | 4% | 0/101 | 0% | - | - | | - | - | |
| **College degree or higher** | 67/143 | 47% | 36/101 | 36% | 0.63 | 0.37–1.06 | .080 | | | |
| **Lower income level (less than minimum wage)** | 43/134 | 32% | 46/94 | 49% | 2.03 | 1.18–3.49 | .010 | | | |
| **Region of birth†** | | | | | | | .001 | | | |
| Europe, the Netherlands | 84/143 | 59% | 47/101 | 57% | Ref | | | | | |
| Europe, other than the Netherlands | 23/143 | 16% | 10/101 | 10% | 0.78 | 0.34–1.77 | | | | |
| Sub-Saharan Africa | 7/143 | 5% | 22/101 | 22% | 5.61 | 2.23–14.13 | | | | |
| Latin America / Caribbean | 15/143 | 10% | 11/101 | 11% | 1.31 | 0.56–3.08 | | | | |
| Other | 14/143 | 10% | 11/101 | 11% | 1.40 | 0.59–3.34 | | | | |
| **Self-defined ethnicity†** | | | | | | | .008 | | | |
| European | 92/141 | 65% | 54/101 | 54% | Ref | | | | | |
| African | 9/141 | 6% | 22/101 | 22% | 4.16 | 1.79–9.70 | | | | |
| American | 4/141 | 3% | 2/101 | 2% | 0.85 | 0.15–4.81 | | | | |
| Asian | 9/141 | 6% | 8/101 | 8% | 1.51 | 0.55–4.16 | | | | |
| Mixed | 13/141 | 9% | 9/101 | 9% | 1.18 | 0.47–2.94 | | | | |
| Latin America / Caribbean | 11/141 | 8% | 2/101 | 2% | 0.31 | 0.07–1.45 | | | | |
| Middle Eastern | 3/141 | 2% | 4/101 | 4% | 2.27 | 0.49–10.53 | | | | |
| **Attending religious services at least once a year** | 42/138 | 30% | 32/99 | 32% | 1.09 | 0.63–1.90 | .757 | | | |

95%-CI, 95% confidence interval; aOR, adjusted odds ratio; OR, odds ratio; IQR, interquartile range; NA, Not applicable.

* Late HIV diagnoses is defined as having had an AIDS-defining illness or a CD4 count <350 cells/mm3 at time of HIV diagnosis.

† Not included in the multivariable model due to collinearity with variable on sexual orientation according to migration status.

## Results

In total, 417 individuals diagnosed with HIV in the preceding 5 years were invited for study participation, of whom 252 (60%) agreed to participate. Eight participants were excluded from further analyses, as stage of HIV-infection at time of diagnosis could not be determined. Among the included 244 participants, 143 (59%) were classified as being diagnosed early and 101 (41%) as late. Of these, respectively 123/143 (86%) and 63/101 (62%) were identified as MSM, and 59/143 (41%) and 54/101 (53%) as migrants (Table 1). Those diagnosed late were older at time of study participation and HIV diagnosis compared to those diagnosed earlier ($p<0.001$ for both). Education level did not differ significantly between groups: 47% of participants with early and 36% of participants with late diagnosis reported to be highly educated ($p = 0.08$).

In multivariable analysis (Table 1), late diagnosis was associated with older age at time of HIV diagnosis (adjusted odds ratio [aOR] = 1.04 per year, 95%-CI = 1.01–1.06). Also, being

heterosexual was associated with late diagnoses: compared to non-migrant MSM, the adjusted odds of being diagnosed late was 1.34 (95%-CI = 0.70–2.57) for migrant MSM, 5.57 (95%-CI = 1.60–19.40) for non-migrant heterosexual males, 11.14 (95%-CI = 3.04–40.81) for migrant heterosexual males, and 3.20 (95%-CI = 1.16–8.88) for migrant heterosexual females ($p<0.001$).

## HIV diagnosis and HIV/STI-testing preceding HIV diagnosis

The majority of migrant participants were diagnosed with HIV in the Netherlands (93% of those diagnosed early and 91% of those diagnosed late; $p = 0.87$; Table 2). Years between migration to the Netherlands and HIV diagnosis did not differ between those diagnosed early or late (median 8 [IQR = 2–24] vs. 6 [IQR = 1–14] years; $p = 0.17$).

Before HIV diagnosis, 89% of participants diagnosed early and 62% who were diagnosed late had previously been tested for HIV-infection ($p<0.001$). Among those previously tested, median time between last negative HIV test and HIV diagnosis was 1 [IQR = 0–3] year for participants diagnosed early and 4 [IQR = 2–9] years for participants diagnosed late ($p<0.001$). In total, 67% of participants diagnosed early and 39% of participants diagnosed late had been diagnosed with a STI prior to their HIV diagnosis ($p<0.001$). Preceding HIV diagnosis, HCV was also more frequently diagnosed in participants with early diagnosis (13% vs. 4%, $p = 0.014$).

## Access to healthcare preceding HIV diagnosis

Difficulty accessing healthcare preceding HIV diagnosis was reported by 9% of participants with early and 13% of participants with late diagnosis ($p = 0.33$) (Table 2). Also when analyses were restricted to migrants only, there was no difference in experienced difficulty accessing healthcare between groups (17% vs. 23%, $p = 0.45$). Healthcare use in the two years preceding HIV diagnosis was reported by the majority whether diagnosed early or late (93% vs. 92%, respectively; $p = 0.79$). The GP and dentist were the most frequently visited healthcare professionals. Those with early diagnosis more frequently visited a sexual health clinic (SHC) preceding diagnosis compared to those with late diagnosis (45% vs. 11%, respectively; $p<0.001$). During SHC visits, HIV testing was discussed with 78% of participants with early and 44% of participants with late diagnosis ($p = 0.048$). There was no difference between groups in the extent to which HIV testing was discussed during visits at other healthcare locations.

## Discussion

In this study, late diagnosis of HIV was associated with older age and being heterosexual in both migrants and non-migrants living with HIV in the Netherlands. We additionally show that preceding HIV diagnosis, a relatively large proportion of people diagnosed late had previously tested for HIV and had high levels of healthcare usage, suggesting missed opportunities for early HIV diagnosis in this group.

Our association between late HIV diagnosis and older age and being heterosexual is in line with previous studies conducted in the Netherlands and other high-income countries [10, 11]. Several other studies additionally showed that late diagnosis is associated with a low HIV risk perception, which was not measured in our study, and being born in a country outside the one of current residence [12]. We previously demonstrated that disparities in access to and use of HIV-related health services and experiences exist by migrant status but also by sexual orientation [7].

Although late stage HIV-infections were associated with being heterosexual, it must be noted that the majority of people diagnosed late in the current study were MSM, reflecting the

**Table 2.** HIV-diagnosis characteristics, HIV/STI-testing and access to healthcare preceding HIV-diagnosis among aMASE-study participants in the Netherlands, 2013–2015, stratified by early versus late HIV presenters.

| | Early diagnosis* (N = 143) | | Late diagnosis* (N = 101) | | p-value |
|---|---|---|---|---|---|
| | n/N | % | n/N | % | |
| *HIV-diagnosis* | | | | | |
| Years since HIV-diagnosis (median, IQR) | 2 | [1–3] | 2 | [1–4] | .070 |
| Location of HIV-diagnosis | | | | | < .001 |
| Sexual health clinic / HIV testing clinic | 80/139 | 58% | 25/96 | 26% | |
| Hospital | 22/139 | 16% | 40/96 | 42% | |
| GP | 30/139 | 22% | 26/96 | 27% | |
| Other[†] | 7/139 | 5% | 5/96 | 5% | |
| Years between migration to the Netherlands and HIV diagnosis (median, IQR)[‡] | 8 | [2–24] | 6 | [1–14] | .168 |
| Country of HIV diagnosis[‡] | | | | | .874 |
| The Netherlands | 55/59 | 93% | 49/54 | 91% | |
| Country of birth | 3/59 | 5% | 4/54 | 7% | |
| Other country | 1/59 | 2% | 1/54 | 2% | |
| *HIV and STI-testing preceding HIV-diagnosis* | | | | | |
| Ever had a negative HIV-test before HIV-diagnosis | 127/143 | 89% | 63/101 | 62% | < .001 |
| Years between previous negative HIV-test and HIV-diagnosis (median, IQR)[ᵞ] | 1 | [0–3] | 4 | [2–9] | < .001 |
| Country of previous HIV negative test [‡,ᵞ] | | | | | .112 |
| The Netherlands | 33/51 | 65% | 14/30 | 47% | |
| Another country | 18/51 | 35% | 16/30 | 53% | |
| Ever had a positive HBV test before HIV diagnosis | 22/143 | 15% | 13/101 | 13% | .581 |
| Ever had a positive HCV test before HIV diagnosis | 19/143 | 13% | 4/101 | 4% | .014 |
| Ever had a STI before HIV diagnosis | 96/143 | 67% | 39/101 | 39% | < .001 |
| *Access to healthcare preceding HIV-diagnosis* | | | | | |
| Registered at a GP in the Netherlands | 141/143 | 99% | 98/101 | 97% | .393 |
| Healthcare usage in the Netherlands two years before HIV-diagnosis[Ψ] | 121/130 | 93% | 78/85 | 92% | .793 |
| Antenatal care | 1/130 | 1% | 0/85 | 0% | .999 |
| Dentist | 90/130 | 69% | 53/85 | 62% | .296 |
| Drug treatment center | 1/130 | 1% | 2/85 | 2% | .564 |
| General practitioner | 89/130 | 69% | 61/85 | 72% | .606 |
| Hospital, emergency room | 24/130 | 19% | 12/85 | 14% | .404 |
| Hospital, inpatient admission | 17/130 | 13% | 17/85 | 20% | .174 |
| Hospital, outpatient clinic | 36/130 | 28% | 14/85 | 17% | .057 |
| Medical care at refugee center | 0/130 | 0% | 1/85 | 1% | .215 |
| Mental health facility | 14/130 | 11% | 5/85 | 6% | .217 |
| Sexual health clinic or HIV testing clinic | 59/130 | 45% | 9/85 | 11% | < .001 |
| HIV-testing discussed during healthcare attendance in the two years before HIV-diagnosis[ₓ] | 61/118 | 52% | 19/75 | 25% | < .001 |
| Antenatal care | 0/1 | 0% | 0/0 | NA | NA |
| Dentist | 2/90 | 2% | 1/53 | 2% | .999 |
| Drug treatment center | 1/1 | 100% | 2/2 | 100% | NA |
| General practitioner | 26/89 | 29% | 12/61 | 20% | .187 |
| Hospital, emergency room | 1/24 | 4% | 0/12 | 0% | .999 |
| Hospital, inpatient admission | 1/17 | 6% | 1/17 | 6% | .999 |
| Hospital, outpatient clinic | 6/36 | 17% | 3/14 | 21% | .697 |
| Medical care at refugee center | 0/0 | NA | 0/1 | 0% | NA |
| Mental health facility | 3/14 | 21% | 0/5 | 0% | .530 |

*(Continued)*

**Table 2.** (Continued)

| | Early diagnosis* (N = 143) | | Late diagnosis* (N = 101) | | *p*-value |
|---|---|---|---|---|---|
| | *n/N* | % | *n/N* | % | |
| Sexual health clinic or HIV testing clinic | 46/59 | 78% | 4/9 | 44% | .048 |
| **Experienced difficulties accessing healthcare in the Netherlands** | 13/143 | 9% | 13/100 | 13% | .332 |

IQR, interquartile range; GP, general practitioner; HBV, hepatitis B virus; HCV, hepatitis C virus; STI, sexually transmittable infection.

* Late HIV diagnosis is defined as having had an AIDS-defining illness or a CD4 count <350 cells/mm3 at time of HIV diagnosis.

† Other includes: antenatal care (n = 3), refugee center (n = 3), fertility clinic (n = 1), dentist (n = 1), self-test (n = 1), medical examination (n = 1), private clinic (n = 1), unknown (n = 1).

‡ Among migrants.

ↄ Only participants were included who had a previous negative HIV test before diagnosis.

Ψ Only participants were included who lived in the Netherlands for two years or more and who were diagnosed with HIV in the Netherlands.

₵ Only participants were included who lived in the Netherlands for two years or more, who were diagnosed with HIV in the Netherlands and who had used healthcare in the Netherlands in the previous two years before HIV diagnosis.

HIV epidemic in many high-income countries including the Netherlands where most newly-diagnosed infections occur among MSM [13]. This suggests that late diagnosis of HIV should be addressed in both groups, and that different strategies are needed to increase timely HIV testing. Our data showing that 62% of participants diagnosed late had previously tested for HIV test is reassuring, as it suggests that most individuals are not averse to HIV testing per se. Ways should be found to encourage more frequent HIV testing among these individuals, as time between last negative HIV test and HIV diagnosis was relatively long in the current study. This might be achieved by discussing the importance of routine HIV testing with individuals whenever they undergo an HIV test.

Our findings additionally support the need to increase healthcare provider-initiated testing, as recommended by the European Centre for Disease Prevention and Control [14]. In our study sample, 92% of participants diagnosed late visited a healthcare provider in the two years preceding HIV diagnosis, in which an HIV test was not often discussed. Efforts have been made in recent years to increase pro-active testing by GPs across Europe, which focus on testing based on indicator disease, sexual orientation, migration background and residence area [15, 16]. The impact of such interventions on testing behavior of GPs is mixed [17–19]. This might be attributable to several wide-spread barriers among GPs including competing priorities in general practice, difficulties with the organizational implementation of HIV testing, fear to discuss or offer HIV testing to patients, lack of registration of sexual orientation and migration background in patient files, and concerns regarding the impact that an HIV test can have on patients [16, 20]. These barriers should be addressed, and other opportunities within the existing healthcare system should be explored. For example, participants of the current study also frequently visited a dentist and were admitted to a hospital preceding HIV diagnosis. Providing HIV testing in these settings are likely hindered by similar barriers as described for GPs, although the majority of dentists in several high-income countries seem willing to conduct rapid HIV tests in their practice [21], which could be useful in cases of oral candidiasis or oral hairy leukoplakia, and intervention to increase targeted HIV testing in hospitals, especially emergency rooms, show promising results in some settings [22, 23].

A limitation of the current study was the lack of detailed information on motives and barriers for HIV testing among our participants. This limited us from exploring reasons why HIV testing was not performed earlier among people diagnosed late. These issues will need greater

consideration in future studies. Furthermore, healthcare systems and characteristics of risk populations might differ across countries. Our results should therefore be generalized to other countries with caution. Finally, the generalizability of our study might be subject to selection bias, as 40% of individuals invited for study participation refused to participate. A previous study using aMASE data from Dutch recruitment sites showed that especially migrants from Latin America/Caribbean, non-migrant women and heterosexual men were less likely to participate, and therefore are likely underrepresented in the current study [7].

In conclusion, interventions to increase HIV testing are needed to diagnose and treat HIV-infections in an early stage. As most people diagnosed late used healthcare preceding their HIV diagnosis in which HIV testing was not discussed, it seems feasible to successfully roll out interventions to increase testing within the existing healthcare system. Simultaneously, efforts should be made to encourage future repeated or routine HIV testing among individuals whenever they undergo an HIV test.

## Supporting information

**S1 Appendix. aMASE clinic questionnaire, Dutch version.**
(PDF)

**S2 Appendix. aMASE clinic questionnaire, English version.**
(PDF)

## Author Contributions

**Conceptualization:** Ward P. H. van Bilsen, Maria Prins.

**Data curation:** Janneke P. Bil, Jan M. Prins, Kees Brinkman, Eliane Leyten, Ard van Sighem.

**Formal analysis:** Ward P. H. van Bilsen.

**Methodology:** Janneke P. Bil.

**Project administration:** Ard van Sighem, Fiona Burns, Maria Prins.

**Supervision:** Maria Prins.

**Writing – original draft:** Ward P. H. van Bilsen.

**Writing – review & editing:** Janneke P. Bil, Jan M. Prins, Kees Brinkman, Eliane Leyten, Ard van Sighem, Maarten Bedert, Udi Davidovich, Fiona Burns, Maria Prins.

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
