## [Decision Letter · Decision Letter 0]

26 Oct 2021

PONE-D-21-18622Testing and healthcare seeking behavior preceding HIV diagnosis among migrant and non-migrant individuals living in the Netherlands: directions for early-case findingPLOS ONE

Dear Dr. van Bilsen,

Thank you for submitting your manuscript to PLOS ONE. After careful consideration, we feel that it has merit but does not fully meet PLOS ONE’s publication criteria as it currently stands. Therefore, we invite you to submit a revised version of the manuscript that addresses the points raised during the review process.

We look forward to receiving your revised manuscript.

Kind regards,

Justyna Dominika Kowalska

Academic Editor

PLOS ONE

Journal Requirements:

2. Please include additional information regarding the survey or questionnaire used in the study and ensure that you have provided sufficient details that others could replicate the analyses. For instance, if you developed a questionnaire as part of this study and it is not under a copyright more restrictive than CC-BY, please include a copy, in both the original language and English, as Supporting Information. In addition, please include any details concerning the validation of this tool within the Methods.

Reviewers' comments:

Reviewer's Responses to Questions

**Comments to the Author**

1. Is the manuscript technically sound, and do the data support the conclusions?

Reviewer #1: Yes

Reviewer #2: Yes

2. Has the statistical analysis been performed appropriately and rigorously? 

Reviewer #1: Yes

Reviewer #2: Yes

3. Have the authors made all data underlying the findings in their manuscript fully available?

Reviewer #1: Yes

Reviewer #2: Yes

4. Is the manuscript presented in an intelligible fashion and written in standard English?

Reviewer #1: Yes

Reviewer #2: Yes

5. Review Comments to the Author

Reviewer #1: The presented analysis touches on a very important topic in Europe Testing and healthcare seeking behaviour preceding HIV diagnosis among migrant and non-migrant individuals living in the Netherlands: directions for early-case finding.

The prepared material lacks the broader context of the situation described in the INTRODUCTION. There is no information about the epidemiological situation. There is even a lack of basic information on the number of infected people per year of tests, the proportion of HIV migrants and the specificity of HIV infection in the country described. Therefore, it is difficult to relate the obtained results to the general situation in the described country.

There is also a lack of information about the testing system, is it open to the public, paid / free of charge, do the migrants have the same access for testing as other residents? Is there any barriers: eg language, knowledge.

The METHODS: There is no explanation why 90% of the study was male? Do they test themselves more often, why is there such a small percentage of women both on tests and among migrants? Is it a national specificity or a deliberate selection for the study. If so, what was the decision made.

40% of the respondents refused to take part in the survey, there is no comment on this relatively large percentage

The DISSCUSION part also needs improvement. There are few references to research on similar topics. Whether the obtained results are consistent with the previous research on testing, or are they different in terms of characteristics.

It is worth comparing the obtained results to other studies on testing among migrants in Europe.

The whole article is very interesting, but actually how it was written was the lack of detailed information on motives and barriers for HIV testing among our participants. This could be an interesting issue for further analysis.

Reviewer #2: On my mind the objectives of this article is extremely important, because late HIV diagnostic remains as a huge problem not only for Western Europe but for Eastern Europe as well, where I am from.

Cross-sectional study design and statistical analysis using multivariable logistic regression were good conducted. Previously, I did not find the criteria inclusion for late diagnosis in methods. But I found them in Table 1: Late HIV diagnoses is defined as having had an AIDS-defining illness or a CD4 less than 350 cells/mm3 at time of HIV diagnosis. I think it is better include to methods. Manuscript presented in an intelligible fashion and written in standard English. Authors made all data underlying the findings in their manuscript fully available.

In conclusion, interventions to increase HIV testing are needed to diagnose and treat HIV-infections in an early stage. As most people diagnosed late used healthcare preceding their HIV diagnosis in which HIV testing was not discussed, it seems feasible to successfully roll out interventions to increase testing within the existing healthcare system. Simultaneously, efforts should be made to encourage future repeated or routine HIV testing among individuals whenever they undergo an HIV test.

6. PLOS authors have the option to publish the peer review history of their article (what does this mean?). If published, this will include your full peer review and any attached files.

Reviewer #1: **Yes: **Magdalena Ankiersztejn-Bartczak

Reviewer #2: **Yes: **Tetiana Kyrychenko

---

## [Author Response · Author response to Decision Letter 0]

12 Jan 2022

Reviewers’ comments:

Reviewer #1 

The presented analysis touches on a very important topic in Europe Testing and healthcare seeking behaviour preceding HIV diagnosis among migrant and non-migrant individuals living in the Netherlands: directions for early-case finding.

The prepared material lacks the broader context of the situation described in the INTRODUCTION. There is no information about the epidemiological situation. There is even a lack of basic information on the number of infected people per year of tests, the proportion of HIV migrants and the specificity of HIV infection in the country described. Therefore, it is difficult to relate the obtained results to the general situation in the described country.

There is also a lack of information about the testing system, is it open to the public, paid / free of charge, do the migrants have the same access for testing as other residents? Is there any barriers: eg language, knowledge.

Response: We acknowledge that the introduction was fairly brief and that additional information on the HIV epidemiology and testing services in the Netherlands is useful for the interpretation of our data. In the introduction section of the revised manuscript, we therefore elaborated on these topics. 

The METHODS: There is no explanation why 90% of the study was male? Do they test themselves more often, why is there such a small percentage of women both on tests and among migrants? Is it a national specificity or a deliberate selection for the study. If so, what was the decision made.

40% of the respondents refused to take part in the survey, there is no comment on this relatively large percentage

Response: In the Netherlands, most new HIV infections occur in men who have sex with men (61% in 2019). Also in our study, the majority of participants were MSM (87%), which resulted in a relatively high proportion of men in the study. In table 1, the proportion of males in our study is specified, as well as the proportion of MSM and heterosexual individuals. 

With regard to the response rate, we added a comment on this in the limitation section of the revised manuscript. 

The DISSCUSION part also needs improvement. There are few references to research on similar topics. Whether the obtained results are consistent with the previous research on testing, or are they different in terms of characteristics.

It is worth comparing the obtained results to other studies on testing among migrants in Europe.

The whole article is very interesting, but actually how it was written was the lack of detailed information on motives and barriers for HIV testing among our participants. This could be an interesting issue for further analysis.

Response: We agree with the reviewer that our discussion section lacked comparison between our study findings and that of previous studies or data of other high-income countries. In the revised version of our manuscript, we added information on this topic. We moreover mention that it would be interesting to further investigate motives and barriers for HIV testing among those with a late HIV diagnosis. 

Reviewer #2

On my mind the objectives of this article is extremely important, because late HIV diagnostic remains as a huge problem not only for Western Europe but for Eastern Europe as well, where I am from.

Cross-sectional study design and statistical analysis using multivariable logistic regression were good conducted. Previously, I did not find the criteria inclusion for late diagnosis in methods. But I found them in Table 1: Late HIV diagnoses is defined as having had an AIDS-defining illness or a CD4 less than 350 cells/mm3 at time of HIV diagnosis. I think it is better include to methods. Manuscript presented in an intelligible fashion and written in standard English. Authors made all data underlying the findings in their manuscript fully available.

In conclusion, interventions to increase HIV testing are needed to diagnose and treat HIV-infections in an early stage. As most people diagnosed late used healthcare preceding their HIV diagnosis in which HIV testing was not discussed, it seems feasible to successfully roll out interventions to increase testing within the existing healthcare system. Simultaneously, efforts should be made to encourage future repeated or routine HIV testing among individuals whenever they undergo an HIV test.

Response: We thank the reviewer for this feedback. As suggested, we added the definition of late HIV diagnosis to the Method section of the revised manuscript.

---

## [Decision Letter · Decision Letter 1]

11 Feb 2022

Testing and healthcare seeking behavior preceding HIV diagnosis among migrant and non-migrant individuals living in the Netherlands: directions for early-case finding

PONE-D-21-18622R1

Dear Dr. van Bilsen,

We’re pleased to inform you that your manuscript has been judged scientifically suitable for publication and will be formally accepted for publication once it meets all outstanding technical requirements.

Kind regards,

Awachana Jiamsakul, PhD

Academic Editor

PLOS ONE

Additional Editor Comments (optional):

Reviewers' comments:

Reviewer's Responses to Questions

**Comments to the Author**

1. If the authors have adequately addressed your comments raised in a previous round of review and you feel that this manuscript is now acceptable for publication, you may indicate that here to bypass the “Comments to the Author” section, enter your conflict of interest statement in the “Confidential to Editor” section, and submit your "Accept" recommendation.

Reviewer #1: All comments have been addressed

Reviewer #2: (No Response)

2. Is the manuscript technically sound, and do the data support the conclusions?

Reviewer #1: Yes

Reviewer #2: Yes

3. Has the statistical analysis been performed appropriately and rigorously? 

Reviewer #1: Yes

Reviewer #2: Yes

4. Have the authors made all data underlying the findings in their manuscript fully available?

Reviewer #1: Yes

Reviewer #2: Yes

5. Is the manuscript presented in an intelligible fashion and written in standard English?

Reviewer #1: Yes

Reviewer #2: Yes

6. Review Comments to the Author

Reviewer #1: The article deals with a very important problem in Europe. The comments made in the first review have been incorporated in the revised version of the article. The additions added by the authors help to better understand the context of the HIV epidemic in the country.

Reviewer #2: I am very satisfied with results of the research and findings that late diagnosis was significantly associated with older age and being heterosexual.

It very interesting that a large proportion of people diagnosed late had previously tested for HIV and had high levels of healthcare usage. Really, efforts should be made to encourage future repeated or routine HIV testing among individuals whenever they undergo an HIV test. In Ukraine we have late diagnosis of HIV, some patients were diagnosed with HIV more than15 years and they did not visit doctor until feel worse. I think we can collaborate for future research. Please, feel free to send me your proposal on email tanakyrychenko@gmail.com

7. PLOS authors have the option to publish the peer review history of their article (what does this mean?). If published, this will include your full peer review and any attached files.

Reviewer #1: **Yes: **Magdalena Ankiersztejn-Bartczak

Reviewer #2: **Yes: **Tetiana Kyrychenko

---

## [Editor Report · Acceptance letter]

22 Feb 2022

PONE-D-21-18622R1 

Testing and healthcare seeking behavior preceding HIV diagnosis among migrant and non-migrant individuals living in the Netherlands: directions for early-case finding 

Dear Dr. van Bilsen:

I'm pleased to inform you that your manuscript has been deemed suitable for publication in PLOS ONE. Congratulations! Your manuscript is now with our production department. 

Kind regards, 

on behalf of

Dr. Awachana Jiamsakul 

Academic Editor

PLOS ONE